# 40-Year Projections of Disability and Social Isolation of Older Adults for Long-Range Policy Planning in Singapore

**DOI:** 10.3390/ijerph17144950

**Published:** 2020-07-09

**Authors:** Reuben Ng, Si Qi Lim, Su Ying Saw, Kelvin Bryan Tan

**Affiliations:** 1Lee Kuan Yew School of Public Policy, National University of Singapore, Singapore 259772, Singapore; lim.siqi@nus.edu.sg; 2Lloyds Register Foundation Institute for the Public Understanding of Risk, National University of Singapore, Singapore 117602, Singapore; 3Ministry of Health, Singapore 169854, Singapore; usprng@nus.edu.sg (S.Y.S.); gmsncwr@nus.edu.sg (K.B.T.)

**Keywords:** aging, social gerontology, Asia, risk, psychomics, public policy

## Abstract

Against a rapidly aging population, projections are done to size up the demand for long-term care (LTC) services for long-range policy planning. These projections are typically focused on functional factors such as disability. Recent studies indicate the importance of social factors, for example, socially isolated seniors living alone are more likely to be institutionalized, resulting in higher demand for LTC services. This is one the first known studies to complete a 40-year projection of LTC demand based on disability and social isolation. The primary micro dataset was the Retirement and Health Survey, Singapore’s first nationally representative longitudinal study of noninstitutionalized older adults aged 45 to 85 with over 15,000 respondents. Disability prevalence across the mild to severe spectrum is projected to increase five-fold over the next 40 years, and the number of socially isolated elders living alone is projected to grow four-fold. Regression models of living arrangements revealed interesting ethnic differences: Malay elders are 2.6 times less likely to live alone than their Chinese counterparts, controlling for marital status, age, and housing type. These projections provide a glimpse of the growing demand for LTC services for a rapidly aging Singapore and underscore the need to shore up community-based resources to enable seniors to age-in-place.

## 1. Introduction

Projections are integral to long-term policy formulation. Traditionally, projections rely on macro indicators (e.g., GDP) to provide big picture directions, although they are too generic for detailed planning. In a rapidly aging country like Singapore, and a civil service striving for precision interventions, more granular projections are required. We report the innovative use of microdata (e.g., surveys) to project disability prevalence and social isolation among older adults.

Our study has both policy and conceptual significance: From a policy perspective, such granular projections are more useful for long-term care and community care planning [1]. Conceptually, this is one of the first known studies to project long term care (LTC) demand based on both functional (disability) and social factors (social isolation). Previous studies relied narrowly on functional variables (disability) to determine future LTC needs. Before explaining our projections, we describe the global aging context and synthesize the extant literature on the factors-functional and social-that contribute to demand for LTC services.

The world’s aging population will experience unprecedented growth: Seniors 60 years and above will grow from 610 million in 2000 to over two billion by 2040 [2]. Asia will constitute over half of the world’s oldest old persons by 2030 [3]. In 2050, over 60% of Asia’s population will be above 60 years, and more than half of this group will be over 80 years. According to a United Nations report, the fastest aging population belongs to Northern Africa and Western Asia, from 29 million in 2019 to 96 million in 2050 [4]. Over the next 30 years in Australia, musculoskeletal and cardiovascular-related diseases are projected to make up 70% of profound or severe core activity restriction (i.e., limited mobility, speech, care) in older Australians [3]. By 2026, the National Disability Insurance Scheme (NDIS) of Australia estimated that Australians aged 65 years and over will increase to over 100,000 and will be required to carry out vital support planning and costing of the scheme [5]. In England and Wales, a study found that life expectancy will expand, and the increasing care needs for older people will rise by 25% in 2025. Particularly, people aged 65 years and over will develop some form of disability [6].

These global statistics reflect the inexorable demand for long-term care (LTC) services and the need to plan strategically. Traditionally, demand for LTC services is driven by functional factors (e.g., disability), although recent studies indicate the importance of social factors [7]. For example, seniors living alone are more likely to be institutionalized, resulting in higher demand for LTC services [8]. Against this background, the paper provides projections of disability prevalence and social isolation to enable Singapore to right-size the demand for long-term policy planning. We draw primarily from the Retirement and Health Study (RHS), Singapore’s first nationally representative longitudinal study of over 15,000 older adults.

### Functional and Social Drivers of Demand for LTC Services

Functional decline is one of the greatest threats to independence in older adults and leads to varying disability in performing activities of daily living (ADL) [9]. There are three categories of disability-mild (1–2 ADLs), moderate (3–4 ADLs), and severe (5–6 ADLs)-that correspond to the progressive inability to perform various ADLs [10]. Cognitive decline, though distinct, is related to physical decline by hastening the latter. For example, individuals with dementia have a higher likelihood of disability [9]. Increasing deficits result in higher health care utilization and expenditure. Functional decline is the key determinant for subsidies. In Singapore, the ElderShield Disability Insurance Scheme, launched in 2002, provides monthly payouts to disabled adults from the age of 40. Older adults (born before 1950) can claim benefits under the Pioneer Generation Disability Assistance Scheme. The key eligibility criterion is the inability to perform three ADLs. To align with the Sustainable Development Goals (SDGs) in establishing the economic inclusion of persons with disabilities, Singapore introduced a new Home Caregiving Grant (HCG) in 2019. It took into account the caregivers, regardless of age, responsible for caring for a person with a permanent moderate disability, by providing a SGD 200 monthly grant to partially offset the cost of hiring a foreign domestic worker [11].

Beyond functional drivers, there are social drivers of demand for LTC services. Social drivers, such as isolation, are crucial determinants of the social and financial position of older adults, and the accessibility to social support arrangements [12]. The World Health Organization (WHO) identified social isolation’s association with an increased risk of institutionalization. Specifically, disabled elders who live alone tend to require increased need in paid care services compared to those who live with children or a spouse [13]. Several studies have converged on the negative effects of living alone, and there is emerging evidence that living with company is a better alternative [14]. Against this background, we ordered living arrangements, with increasing efficacy, through the following: (1) living alone; (2) living with spouse; (3) living with others; (4) living with children or grandchildren, predicated on the following evidence.

Numerous studies proposed that living alone tends to be the least efficacious: social isolation is linked to decreased happiness and increased depression [15]. Across ethnicity and gender, the number of older adults living alone has increased, and the highest prevalence of social isolation is found in the UK and Finland [12,16]. On the other hand, living with a spouse achieves reciprocal support, provides emotional succor, and long-term mutual commitment to interests and values [17]. In addition, elders who live with others displayed better health outcomes because of the benefits from social support. There is substantial evidence to suggest that social relationships, exemplified by living with others, are linked to lower mortality risk and increase mental wellness [18]. Specifically, living with children/grandchildren achieves the most optimal mental and physical health outcomes. With regards to mental health, participation in the caregiving of grandchildren provides meaning and purpose [15]. Against this background, living arrangements can be ranked based on efficacy to an elder’s well-being, with social isolation (living alone) as the least efficacious to living with children/grandchildren as the most efficacious.

On a separate note, while previous studies have explored the differences in living arrangements across west and east, no known studies have investigated how ethnic diversity within eastern cultures influence living arrangements [3]. In addition to other contributions, our study closes this gap by analyzing how living arrangements differ across Asian ethnicities: Chinese, Malays and Indians.

## 2. Materials and Methods

### Dataset and Analytic Strategy

The primary dataset is the Retirement and Health Study (RHS), Singapore’s first and largest nationally representative longitudinal study of noninstitutionalized older adults spanning the age range 45–85 with over 15,000 respondents. Sample demographics are presented in Table 1. Conducted through a face-to-face interview, participants underwent extensive assessments of health, savings, and expenditure by trained interviewers. Given the multiethnic and multilingual nature of the older Singaporeans, the interviews were conducted in the language and/or dialect that each participant was most conversant. Informed consent was sought by the participant or on behalf of their respective legally acceptable representative (LAR) when the participant was bounded with some form of disability (i.e., physical, mental, and cognitive).

Established procedures were used to distil the projections of disability and living arrangements:Disability prevalence for 2010 and 2014 were taken from nationally representative surveys: Social Isolation, Health & Lifestyles study (SIHS) and RHS, respectively.To derive the age-specific proportion of seniors by living arrangements in 2034, the cohort of residents aged 45 and over in 2014 was used to form the senior cohort in 2034. We obtained the predicted living arrangements for this cohort in 2034 through ordered and multinomial logit models that achieve convergent results.

## 3. Results

To reiterate, projections of disability and social isolation are crucial for estimating the demand for long-term care (LTC) services. This section illustrates disability projections and projections of social isolation.

### 3.1. Disability Projections

Among seniors aged 65 and above, comparing 2010 with 2014, moderate and severe disability increased while mild disability decreased. Specifically, moderate disability prevalence inched from 6.09% to 6.64%; severe disability increased from 2.41% to 3.33%; mild disability decreased from 13.12% to 9.63% in Table 2.

By applying the disability prevalence in Table 2 to the population figures from the Department of Statistics (DOS) for residents aged 45 years and older, we obtained the projected disability level estimates from 2020 to 2060. Between 2014 and 2030, we expect the number of seniors with at least mild disability (≥1 ADL) to increase by 2.4 times from 42,000 to 100,000. Within this group, we expect the number of seniors with moderate to severe disability (≥3 ADLs) to increase by 1.4 times from 29,000 to 69,000 during the same period in Figure 1.

### 3.2. Social Isolation Projections

Elders who live alone are more likely to be institutionalized, resulting in higher demand for LTC services. Figure 2 shows the age-specific proportion of seniors by their living arrangements.

We included seniors living in nursing homes from administrative data by taking their living arrangements prior to entering the institution. From the 2014 figures, 9.8% or 51,100 seniors lived alone, and the prevalence rose with age. To distil the predictors of living arrangements among elders, we ran ordered and multinomial logistic regressions on residents aged 65 and above with living arrangement as the outcome where we derived convergent findings.

Results show interesting ethnic differences (Table 3): Malay elders are 2.6 times less likely than Chinese to live alone, after controlling for their marital status, age, and housing type (a proxy for socioeconomic status) in an ordered logit model. Multinomial logit models corroborated these findings: minority ethnic groups evidenced a decreased chance of social isolation. To derive the age-specific proportion of seniors by living arrangements in 2060, we used the cohort of residents aged 45 and above in 2014 since they will form the senior cohort in four decades. We obtained the predicted living arrangements for this cohort, controlling for covariates. Thereafter, we scaled up the figures to take into account the seniors that would be staying in nursing homes assuming that the institutionalized rates remain at 2014 levels. Overall, the number of socially isolated seniors are projected to increase four times over 40 years. This is plausible given that there is a higher proportion of singles and married seniors without children among younger cohorts compared to older cohorts in Figure 3.

## 4. Discussion

Projecting health scenarios that impact society is important to shape health policy [19,20]. This is a useful tool for a rapidly aging society to right-size demand for LTC services in the coming decades. To this end, we projected the 40-year prevalence of disability and social isolation as functional and social drivers of LTC demand, respectively.

Disability prevalence is projected to increase by five times in 40 years, and the number of socially isolated seniors living alone is projected to increase by four times over 40 years. Models of living arrangements found interesting ethnic differences: Malay seniors are 2.6 times less likely to be socially isolated compared to their Chinese counterparts, controlling for marital status, age, and housing type. These projections indicate the scale of demand for LTC services and the increasing burden on caregivers and community resources. Demand for LTC services is becoming more heterogeneous beyond nursing homes [21]. Increasingly, policy makers are expanding the scope of LTC services to home care and community care to promote interventions-in-place and aging-in-place [22]. These findings contribute to a more granular and realistic calibration of LTC demand and adds to the growing literature on the impact of an aging population from North Africa, Australia, and the UK [2,3,6].

Singapore launched the Community Networks for Seniors (CNS) program to encourage active aging and social connectedness among socially isolated seniors [23]. This was given national prominence when Singapore’s Minister for Finance spoke extensively about CNS during his 2018 annual budget speech in Parliament [24]. Spearheaded by the Ministry of Health (MOH) and its Agency for Integrated Care (AIC), the CNS uses different stakeholders in the community-voluntary welfare organizations, grassroots organizations, regional health systems, and government agencies-to jointly engage and support our seniors. CNS has successfully activated more than 70 Residents’ Committees (RCs) in conducting regular preventive health and active aging activities for more than 70,000 seniors. More than 1500 seniors are now attending these activities on a weekly basis. CNS has also matched more than 600 seniors to befrienders and assisted about 800 seniors with complex health and social needs. Importantly, our projections provide insights into how much and how fast to scale CNS.

In Boston, a ‘village model’, developed and governed by community-dwelling older adults, aimed at promoting social inclusion through a combination of bonding and social networking [25]. For instance, they devised programs targeted for social engagement and volunteerism, from discounted services to home healthcare access and local leadership activities. This is a promising model for aging-in-place, especially in the empowerment of older adults when developing intervention strategies or programs, and they serve as catalysts to promote social inclusion.

A future iteration of the projections could use admin data from the National Electronic Medical Records (NEMR) to mitigate the widely known limitations of self-reported survey data [26]. Beyond projections, simulations can also be done to shed light on how policy changes could affect disability prevalence and acute care [27]. Future studies could also explore the impact of ageism on disability projections as ageism has increased over time [28], and across countries [29,30], as cultural factors impact a variety of outcomes [31,32]. The Covid-19 pandemic will undoubtedly bring unprecedented changes to aging policies, notwithstanding the asymmetrical mortality toll amongst older adults. Future studies should attempt to model the impact of Covid-19 on LTC demand.

## 5. Conclusions

In sum, this is one of the first known studies to project disability prevalence and social isolation. This is useful for policy makers to right-size the demand for LTC services and chart a long-term strategy to manage a rapidly aging population. Of broader significance, this study highlighted the importance of social isolation in projecting the demand for LTC services.

## Figures and Tables

**Figure 1 ijerph-17-04950-f001:**
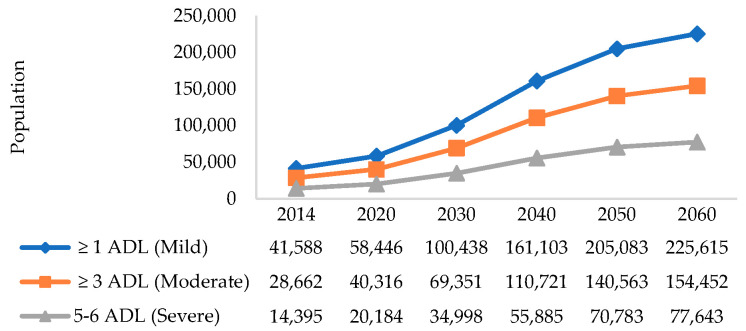
Forty-year projected disability prevalence (mild, moderate, severe) until 2060 in Singapore.

**Figure 2 ijerph-17-04950-f002:**
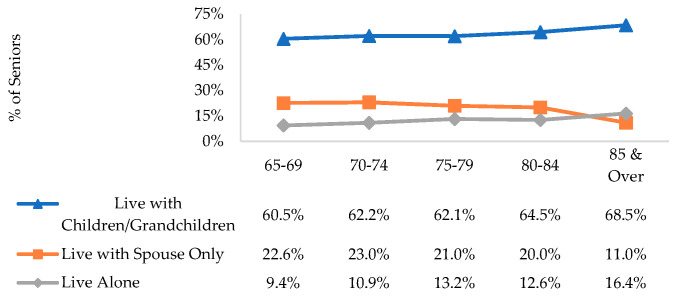
Age-specific proportion of seniors by living arrangements in 2014.

**Figure 3 ijerph-17-04950-f003:**
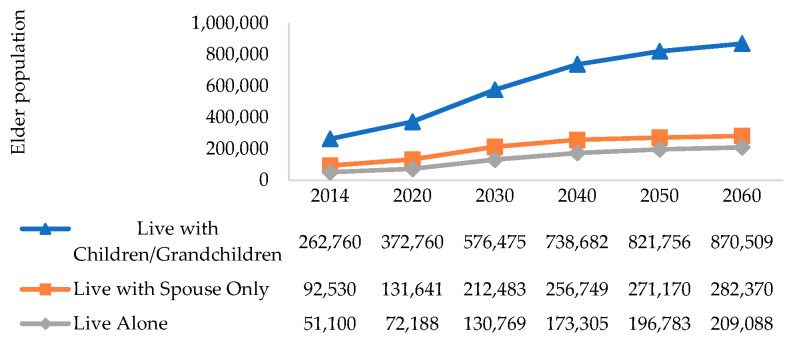
Projected number of elders by living arrangements over 40 years until 2060.

**Table 1 ijerph-17-04950-t001:** Demographic distribution of the sample.

Variable	Proportion ^1^ (%)
Age ^2^	
45–54 years	5183 (41.21%)
55–64 years	4676 (32.96%)
65–74 years	3458 (17.54%)
75–84 years	1684 (7.82%)
≥85 years	102 (0.45%)
Gender	
Male	7861 (50.60%)
Female	7242 (49.40%)
Ethnicity	
Chinese	8073 (74.85%)
Malay	3455 (10.58%)
Indian	2515 (8.45%)
Others	1060 (6.10%)

^1^ Numbers are unweighted, percentages (in parentheses) are weighted and calculated within columns. ^2^ Age as at year of completing the interview.

**Table 2 ijerph-17-04950-t002:** Disability prevalence in 2010 and 2014 by age.

Age Bands	2010			2014		
	≥Mild	≥Mod	Severe	≥Mild	≥Mod	Severe
65–69 years	4.40%	1.81%	1.04%	3.27%	2.14%	0.95%
70–74 years	6.29%	2.95%	1.19%	5.69%	4.22%	2.10%
75–79 years	13.14%	6.06%	2.36%	9.18%	6.38%	3.50%
80–84 years	25.75%	11.44%	3.10%	19.87%	13.68%	6.97%
≥85 years	50.75%	25.19%	10.62%	35.02%	23.70%	11.77%
M	13.12%	6.09%	2.41%	9.63%	6.64%	3.33%

**Table 3 ijerph-17-04950-t003:** Factors associated with living arrangements.

Predictor	Outcome: Living Arrangement
	Odds Ratio
**Ethnicity**	
Chinese	Reference
Malay	2.59 ***
Indian	0.94
Others	0.91
**Marital Status and Number of Children**	
Single	Reference
Married with no Kids	19.76 ***
Married with one Kid	106.79 ***
Married with two or more Kids	131.79 ***
**Age Ranges**	
65–69	Reference
70–74	0.88
75–79	0.85
80–84	0.90
85+	0.92
**Housing Type**	
Public Apartment (1 bedroom)	Reference
Public Apartment (2 bedrooms)	3.22 ***
Public Apartment (3 bedrooms)	6.53 ***
Public Apartment (larger than 3 bedrooms)	9.02 ***
Private Condominiums	4.56 ***
Landed Property	5.99 ***

*** *p* < 0.01.

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
