# Peer review of "40-Year Projections of Disability and Social Isolation of Older Adults for Long-Range Policy Planning in Singapore"

_ijerph, 2020, doi:10.3390/ijerph17144950_

Round 1

Reviewer 1 Report

A very important article in the current context of an again population and shrinking services and political will to cater to vulnerable populations. The article indeed sheds light on the social isolation faced in an important demographic living in the far East. As someone looking into disabilities and vulnerable communities I believe it is very important to present a complete picture of the situation globally and compare it with the situation in Singapore. Merely reading out data without a global comparison is not enough.

Please provide a global picture in the discussion and also highlight in the introduction.

A table comparing demographics of social isolation, institutionalization and other variables would be highly valuable.

DISCUSSION SECTION

Discussion needs to be expanded, it is too brief, followed by key recommendations for practice . I would suggest more data from NHS in the UK and Public health agencies in USA /Australia. This is a an important topic that needs a comparison in the discussion , preferably in the form of a chart ,detailing geographical projections , assessment and assimilating them in a form that is readily available for key policy makers to read and implement in their practice .

CONCLUSION SECTION

Covid19 has taken a hit on the elderly and vulnerable populations across the global, it would be nice to include a section about this in the conclusion.

Below I have put forward a key list of articles that might be useful for the author in creating a comparative analysis.

  1. Piachaud D, Bennett F, Nazroo J, Popay J. Task group submission to the marmot review. 2009. Report of task group 9: social inclusion and social mobility. [Google Scholar]
  2. Lenoir R. Les Exclus: Un Francais sur Dix. 1. Paris: Editions du Seuil; 1974. [Google Scholar]
  3. The World Bank. Social Exclusion and the EU’s Social Inclusion Agenda: Paper Prepared for the EU8 Social Inclusion Study The World Bank. 2007. http://siteresources.worldbank.org/INTECONEVAL/Resources/SocialExclusionReviewDraft.pdf. Accessed 15 Jan 2017.
  4. Peace R. Social exclusion: a concept in need of definition? Soc Policy J N Z. 2001;16:17–36. [Google Scholar]
  5. Daly M. Social Exclusion as Concept and Policy Tempate in the European Union: Center for European Studies Working Paper Series #135: Minda de Gunzburg Center for European Studies Harvard University. 2006. https://ces.fas.harvard.edu/uploads/files/Working-Papers-Archives/CES_WP135.pdf. Accessed 12 Aug 2017.
  6. Atkinson AB. Poverty in Europe. Oxford: Wiley-Blackwell; 1998. [Google Scholar]
  7. Shaw M, Dorling D, Smith GD. Poverty, social exclusion, and minorities. In: Marmot M, Wilkinson R, editors. Social determinants of health. 2. Oxford: Oxford University Press; 2005. pp. 196–223. [Google Scholar]
  8. Popay J, Escorel S, Hernández M, Johnston H, Mathieson J, Rispel L. Understanding and tackling social exclusion. Final report to the WHO Commission on Social Determinants of Health from the Social Exclusion Knowledge Network: World Health Organization. 2008. http://www.who.int/social_determinants/knowledge_networks/final_reports/sekn_final%20report_042008.pdf. Accessed 01 Mar 2016.
  9. Hayes A, Gray M, Edwards B. Social inclusion: origins, concepts and key themes: Australian Institute of Family Studies. 2008. [Google Scholar]
  10. Estivill J. Concepts and Strategies for Combating Social Exclusion: An Overview: International Labour Organization. 2003. http://www.ilo.org/public/english/protection/socsec/step/download/96p1.pdf. Accessed 25 Apr 2017.
  11. The World Bank. Inclusion Matters: The Foundation for Shared Prosperity: The World Bank. 2013. http://documents.worldbank.org/curated/en/114561468154469371/pdf/814780PUB0Incl00Box379838B00PUBLIC0.pdf. Accessed 10 Jan 2017.
  12. Atkinson AB, Marlier E. Analysing and Measuring Social Inclusion in a Global Context: United Nations Department of Economic and Social Affairs. 2010. http://www.un.org/esa/socdev/publications/measuring-social-inclusion.pdf. Accessed 14 Dec 2016.
  13. Labonté RN, Hadi A, Kauffmann XE. Indicators of Social Exclusion and Inclusion: A Critical and Comparative Analysis of the Literature: Institute of Population Health University of Ottawa. 2011. http://www.rrasp-phirn.ca/images/stories/docs/workingpaperseries/wps_dec11_report_final-.pdf. Accessed 12 July 2017.
  14. Mathieson J, Popay J, Enoch E, Escorel S, Hernandez M, Johnston H et al. Social Exclusion Meaning, Measurement and Experience and Links to Health Inequalities: A Review of Literature: WHO Social Exclusion Knowledge Network. 2008. http://www.who.int/social_determinants/media/sekn_meaning_measurement_experience_2008.pdf.pdf. Accessed 01 Feb 2016.
  15. Beall J. Globalization and social exclusion in cities: framing the debate with lessons from Africa and Asia. Environ Urban. 2002;14(1):41–51. [Google Scholar]
  16. Gore C, Figueiredo JB, editors. Social exclusion and anti-poverty policy: a debate. Geneva: International Labour Organization; 1997. [Google Scholar]
  17. Dunn S. Creating Accepting Communities: Report of the Mind Inquiry Into Social Exclusion and Mental Health Problems. London: Mind; 1999. http://www.siis.net/documentos/Digitalizados/900204_Creating%20Accepting%20Communities.pdf. Accessed 01 Apr 2017.
  18. Kelly R. Social Inclusion Policy Community & Enterprise Directorate Clare County Council. 2010. https://www.clarecoco.ie/community/publications/social-inclusion-policy-6116.pdf. Accessed 24 May 2017.
  19. Taket A, Crisp BR, Nevill A, Lamaro G, Graham M, Barter-Godfrey S. Introduction. In: Taket A, Crisp BR, Nevill A, Lamaro G, Graham M, Barter-Godfrey S, editors. Theorising social exclusion. 1. London: Routledge; 2009. [Google Scholar]
  20. Davey S, Gordon S. Definitions of social inclusion and social exclusion: the invisibility of mental illness and the social conditions of participation. Int J Cult Mental Health. 2017;10(3):229–237. [Google Scholar]
  21. Popay J. Understanding and tackling social exclusion. J Res Nurs. 2010;15(4):295–297. [Google Scholar]
  22. Omtzigt DJ. Survey on Social Inclusion: Theory and Policy: Oxford University Institute for Global Economic Development. 2009. http://ec.europa.eu/regional_policy/archive/policy/future/pdf/1_omtzigt_final_formatted.pdf. Accessed 11 Mar 2017.
  23. Topping A. Looting ‘fuelled by social exclusion’. The Guardian. 2011. https://www.theguardian.com/uk/2011/aug/08/looting-fuelled-by-social-exclusion. Accessed 10 Aug 2017.
  24. Borkhataria C. Social exclusion that leads to ‘fake news’ conspiracy theories could be the secret of Donald Trump's success. The Mail Online. 2017. http://www.dailymail.co.uk/sciencetech/article-4243214/Donald-Trump-supporters-felt-socially-excluded.html#ixzz4p9xopA4E. Accessed 05 Aug 2017.
  25. Wilkinson R, Marmot MG, editors. Social determinants of health: the solid facts. 2. Copenhagen: World Health Organization Regional Office for Europe; 2003. [Google Scholar]
  26. World Health Organization Regional Office for Europe. Poverty and Social Exclusion in the WHO European Region: Briefing on policy issues produced through the WHO/European Commission equity project: World Health Organization Regional Office for Europe. 2010. http://www.euro.who.int/__data/assets/pdf_file/0004/127525/e94499.pdf. Accessed 12 Oct 2015.
  27. Gill P, MacLeod U, Lester H, Hegenbarth A. Improving Access to Health Care for Gypsies and Travellers, Homeless People and Sex Workers: Royal College of General Practitioners Clinical Innovation and Research Centre. 2013. http://www.rcgp.org.uk/news/2013/december/~/media/Files/Policy/A-Z-policy/RCGP-Social-Inclusion-Commissioning-Guide.ashx. Accessed 20 Aug 2016.
  28. Cabinet Office Social Exclusion Task Force & Department of Health. Inclusion Health Evidence Pack: Cabinet Office Social Exclusion Task Force and Department of Health. 2010. http://webarchive.nationalarchives.gov.uk/+/http:/www.cabinetoffice.gov.uk/media/346574/inclusion-health-evidencepack.pdf. Accessed 15 Sep 2016.
  29. Department of Health. Inclusion Health: Improving Primary Care for Socially Excluded People: Department of Health. 2010. http://webarchive.nationalarchives.gov.uk/20130124054057/http://www.dh.gov.uk/prod_consum_dh/groups/dh_digitalassets/@dh/@en/@ps/documents/digitalasset/dh_114365.pdf. Accessed 12 May 2016.
  30. World Health Organization. Primary Care: Now More Than Ever: World Health Organization Press. 2008. http://www1.paho.org/hq/dmdocuments/2010/PHC_The_World_Health_Report-2008.pdf. Accessed 01 Mar 2014.
  31. United Nations Department of Economic and Social Affairs. Sustainable development goal 3: ensure healthy lives and promote well-being for all at all ages. United Nations Department of economic and Social Affairs. 2016. https://sustainabledevelopment.un.org/sdg3. Accessed 20 Aug 2017.
  32. Gruskin S, Ferguson L. Using indicators to determine the contribution of human rights to public health efforts. Bull World Health Organ. 2009;87(9):714–719. [PMC free article] [PubMed] [Google Scholar]
  33. Heath I. What is needed. The mystery of general practice. London: Nuffield Provincial Hospitals Trust; 1995. [Google Scholar]
  34. Gill PS, Hegenbarth A. Embedding social inclusion in general practice: time for action. Br J Gen Pract. 2013;63(617):622–623. [PMC free article] [PubMed] [Google Scholar]
  35. Yanicki SM, Kushner KE, Reutter L. Social inclusion/exclusion as matters of social (in)justice: a call for nursing action. Nurs Inq. 2015;22(2):121–133. [PubMed] [Google Scholar]
  36. Le Boutillier C, Croucher A. Social inclusion and mental health. Br J Occup Ther. 2010;73(3):136–139. [Google Scholar]
  37. Levac D, Colquhoun H, O’Brien KK. Scoping studies: advancing the methodology. Implement Sci. 2010;5(1):69. [PMC free article] [PubMed] [Google Scholar]
  38. Arksey H, O’Malley L. Scoping studies: towards a methodological framework. Int J Soc Res Methodol. 2005;8 10.1080/1364557032000119616.
  39. Moher D, Liberati A, Tetzlaff J, Altman DG, Group P Preferred reporting items for systematic reviews and meta-analyses: the PRISMA statement. Open Med. 2009;3(2):123–130. [PMC free article] [PubMed] [Google Scholar]
  40. Williamson M, Allen A. The human givens. Exeter: Mind South West; 2006. [Google Scholar]
  41. Huxley P, Evans S, Munroe M, Webber M, Burchardt T, Knapp M et al. Development of a Social Inclusion Index to capture subjective and objective domains (Phase I): National Co-ordinating Centre for Research and Methodology. 2006. http://citeseerx.ist.psu.edu/viewdoc/download?doi=10.1.1.469.5761&rep=rep1&type=pdf. Accessed 14 Jul 2016.
  42. Australian Mental Health Outcomes and Classification Network (AMHOCN). Development of the Living in the Community (LCQ) measure of social inclusion for use in mental health Final report: Australian Mental Health Outcomes and Classification Network (AMHOCN). 2015. http://www.amhocn.org/sites/default/files/publication_files/living_in_the_community_questionnaire_lcq_final_report.pdf. Accessed 01 Oct 2016.
  43. Coombs T, Nicholas A, Pirkis J. A review of social inclusion measures. Aust N Z J Psychiatry. 2013;47(10):906–919. [PubMed] [Google Scholar]
  44. Chan K, Evans S, Ng Y-L, Chiu MY-L, Huxley PJ. A concept mapping study on social inclusion in Hong Kong. Soc Indic Res. 2014;119(1):121–137. [Google Scholar]
  45. Mezey G, White S, Thachil A, Berg R, Kallumparam S, Nasiruddin O, et al. Development and preliminary validation of a measure of social inclusion for use in people with mental health problems: the SInQUE. Int J Soc Psychiatry. 2012;59(5):501–507. [PubMed] [Google Scholar]
  46. The World Bank. Social development: Sector results profile. The World Bank. 2013. http://www.worldbank.org/en/results/2013/04/14/social-development-results-profile. Accessed 24 Jul 2017.
  47. Marino-Francis F, Worrall-Davies A. Development and validation of a social inclusion questionnaire to evaluate the impact of attending a modernised mental health day service. Ment Health Rev J. 2010;15(1):37–48. [Google Scholar]
  48. Harrison D, Sellers A. Occupation for mental health and social inclusion. Br J Occup Ther. 2008;71(5):216–219. [Google Scholar]
  49. Pachoud B, Plagnol A, Leplege A. Outcome, recovery and return to work in severe mental illnesses. Disabil Rehabil. 2010;32(12):1043–1050. [PubMed] [Google Scholar]
  50. World Health Organization. Promoting Mental Health: Concepts, Emerging Pvidence, Practice: World Health Organization. 2004. http://www.who.int/mental_health/evidence/MH_Promotion_Book.pdf. Accessed 01 May 2017.
  51. Social Exclusion Unit (SEU) Office of the Deputy Prime Minister. Mental Health and Social Exclusion: Social Exclusion Unit Report Summary: Office of the Deputy Prime Minister. 2004. http://www.nfao.org/Useful_Websites/MH_Social_Exclusion_report_summary.pdf. Accessed 23 Nov 2016.
  52. Department of Health and Children. A Vision for Change: Report of the Expert Group on Mental Health Policy: Government of Ireland. 2006. http://www.hse.ie/eng/services/publications/Mentalhealth/Mental_Health_-_A_Vision_for_Change.pdf. Accessed 10 May 2016.
  53. Vigo D, Thornicroft G, Atun R. Estimating the true global burden of mental illness. Lancet Psychiatry. 2016;3(2):171–178. [PubMed] [Google Scholar]
  54. World Health Organization. Mental Health Action Plan 2013–2020: World Health Organization. 2013. http://apps.who.int/iris/bitstream/10665/89966/1/9789241506021_eng.pdf?ua=1. Accessed 13 June 2017.
  55. Vos T, Allen C, Arora M, Barber RM, Bhutta ZA, Brown A, et al. Global, regional, and national incidence, prevalence, and years lived with disability for 310 diseases and injuries, 1990-2015: a systematic analysis for the global burden of disease study 2015. Lancet. 2016;388(10053):1545–1602. [PMC free article] [PubMed] [Google Scholar]
  56. Ramon S, Griffiths CA, Nieminen I, Pedersen M, Dawson I. Towards social inclusion through lifelong learning in mental health: analysis of change in the lives of the EMILIA project service users. Int J Soc Psychiatry. 2009;57(3):211–223. [PubMed] [Google Scholar]
  57. Baumgartner JN, Burns JK. Measuring social inclusion—a key outcome in global mental health. Int J Epidemiol. 2014;43(2):354–364. [PubMed] [Google Scholar]
  58. Adam C, Potvin L. Understanding exclusionary mechanisms at the individual level: a theoretical proposal. Health Promot Int. 2016; 10.1093/heapro/daw005. [PMC free article] [PubMed]
  59. Social Exclusion Unit (SEU) Office of the Deputy Prime Minister. The Social Exclusion Unit: Office of the Deputy Prime Minister. 2004. http://webarchive.nationalarchives.gov.uk/+/http:/www.cabinetoffice.gov.uk/media/cabinetoffice/social_exclusion_task_force/assets/publications_1997_to_2006/seu_leaflet.pdf. Accessed 04 Feb 2017.
  60. Sayce L. From psychiatric patient to citizen: overcoming discrimination and social exclusion. London: Palgrave MacMillan; 2000. [Google Scholar]
  61. Council of the European Union. Joint Report by the Commission and the Council on Social Inclusion: Council of the European Union. 2003. http://ec.europa.eu/employment_social/soc-prot/soc-incl/final_joint_inclusion_report_2003_en.pdf. Accessed 12 July 2017.
  62. Room G. Poverty and social exclusion: the new European agenda for policy and research. In: Room G, editor. Beyond the threshold: the measurement and analysis of social exclusion. Bristol: Policy Press; 1995. [Google Scholar]
  63. Room G, Berghman J, Bouget D, Cabrero G, Hansen F, Hartmann-Hirsch C et al. Observatory on National Policies to Combat Social Exclusion: Second Annual Report: Commission of the European Communities Directorate General Employment, Social Affairs and Industrial Relations. 1992. http://aei.pitt.edu/34932/1/A1082.pdf. Accessed 2 Nov 2016.
  64. Stewart G, Sara G, Harris M, Waghorn G, Hall A, Sivarajasingam S, et al. A brief measure of vocational activity and community participation: development and reliability of the activity and participation questionnaire. Aust N Z J Psychiatry. 2010;44(3):258–266. [PubMed] [Google Scholar]
  65. Berry HL, Rodgers B, Dear KB. Preliminary development and validation of an Australian community participation questionnaire: types of participation and associations with distress in a coastal community. Soc Sci Med. 2007;64(8):1719–1737. [PubMed] [Google Scholar]
  66. Lloyd C, King R, Moore L. Subjective and objective indicators of recovery in severe mental illness: a cross-sectional study. Int J Soc Psychiatry. 2010;56(3):220–229. [PubMed] [Google Scholar]
  67. McColl MA, Davies D, Carlson P, Johnston J, Minnes P. The community integration measure: development and preliminary validation. Arch Phys Med Rehabil. 2001;82(4):429–434. [PubMed] [Google Scholar]
  68. Lloyd C, Waghorn G, Best M, Gemmell S. Reliability of a composite measure of social inclusion for people with psychiatric disabilities. Aust Occup Ther J. 2008;55(1):47–56. [PubMed] [Google Scholar]
  69. Wright N, Stickley T. Concepts of social inclusion, exclusion and mental health: a review of the international literature. J Psychiatr Ment Health Nurs. 2013;20(1):71–81. [PubMed] [Google Scholar]
  70. Waghorn G, Chant D, King R. Classifying socially-valued role functioning among community residents with psychiatric disorders. Am J Psychiatr Rehabil. 2007;10(3):185–221. [Google Scholar]

The article in the current form is incomplete and in my opinion doesn’t add much value to the subject accept for a commentary on the situation in particular country in the world. Once the comparison is complete , it will be a very valuable contribution , it has the potential to be a very valuable contribution if done properly. I would like to see more tables, more analysis and comparison with regional countries and with Eu/UK/US/Australia .

Author Response

Please see detailed responses in the attachment. Thank you.

Reviewer 2 Report

Summary: The current manuscript is aimed to analyze a 40-year projection of long term care demand based on disability and social isolation. Results evidenced that Disability prevalence across the mild to severe spectrum is projected to increase by five-fold over 40 years, and the number of socially isolated elders living alone is projected to grow by four-fold. Moreover, other interesting results are evidenced by risk factor analysis, such as ethnic differences.

I think the study is well conducted, and I have only a few minor comments.  

Specific comments follow:

Introduction: The introduction fits with the goal of the study. Maybe it is useful to distinguish between physical and cognitive decline.

Methods: The method is succinct. Some information should be added about sampling recruitment and could be useful to give more information about respondents.

Results: The summary of the study fits the planned analyses.

Discussion: This section could be improved, emphasizing the novelty of this research. It would be interesting to provide more information on the risk factors highlighted. This, as well as making the results clearer to the reader, could have major clinical implications.

General comment: Generally, I found the study well conducted, although novelty and utility were few emphasized. My advice to the authors is to highlight the innovative nature of the study. This is one of the firsts (presumably the first) study on these fields, and this aspect could be emphasized in discussions. Furthermore, it would be interesting to provide some other implications of results as well as the social implications of this study.

Author Response

(The authors gave the same response as above.)

Reviewer 3 Report

The regression analysis is incorrectly used in this study. The ordered logistic model is obviously not suitable for the analysis.

Normally, the ordered logistic model should be used when the outcome variable has intrinsic rationality to be sequenced. For example, the allocation of pension investment in risky asset (0%, 50%, 100%), or the test score (low, medium, high) etc. usually serve as outcome variables in ordered logistic model.

In this study, the outcome variable (living arrangement) include following categories: living alone, living with spouse only, living with others, and living with children/grandchildren. There is no internal rationality to sequence the outcome variable in a certain order. Can we indicate the rationality why "living with spouse only" should or should not be placed before "living with others". As such, the outcome variable in this study is inappropriate for sequencing. The authors should consider other types of model. 

Author Response

(The authors gave the same response as above.)

Round 2

Reviewer 1 Report

The work after a major overhaul has been greatly improved. The comparison with services present in other countries and the evaluation of recent literature on the subject made the manuscript interesting for readers and increased its scientific value.

A good job has been done

Author Response

Thank you very much.  We are indebted to your important feedback that strengthened the paper. 

Reviewer 3 Report

In the revised manuscript, the critical flaw of this study has still not been adequately addressed. The ordered logit model lacks justification. The empirical results remain highly rely on and sensitive to the specific order/sequence of outcome variable. The slight change of the order would definitely yield a very different result.

The authors propose that living alone implies the least living efficacy and living with children/grand children implies the most, through which to justify why the latter one should be ranked after the former one.

However, the authors still avoid explaining why "living with others" should be ranked after "living with spouse" and before "living with children/grandchildren".

Moreover, what does the word "others" exactly mean in the phrase "living with others"? We can't find any description in the manuscript. Besides, what's the rationale "others" should be ranked after "spouse" and before "children/grandchildren"?

Not all variables are suitable for ordered logit model, unless there is sound justification for the sequence of outcome variable. Otherwise, the empirical results would be highly sensitive to the arbitrary sequence of outcome variable, and fail to yield any meaningful conclusion.

Author Response

Thank you for the valuable feedback.  We agree that “living with others” is a problematic category and have removed it.  We also ran a multinomial logistic regression that corroborated our earlier findings.  These models are described on p3 of the manuscript and and we present the output table from our multinomial logit model in the attached document.  We are indebted to your important recommendations that helped strengthen the paper. 
